# Heuristic Learning as a Method for Improving Students’ Teamwork Skills in Physical Education

**DOI:** 10.3390/ijerph191912596

**Published:** 2022-10-02

**Authors:** Tiziana D’Isanto, Sara Aliberti, Gaetano Altavilla, Giovanni Esposito, Francesca D’Elia

**Affiliations:** 1Department of Human, Philosophical and Education Sciences, University of Salerno, 84084 Fisciano, Italy; 2Departamento de Ciencias de la Actividad Física y del Deporte, Universidad Católica San Antonio de Murcia, 30107 Murcia, Spain; 3Department of Political and Social Studies, University of Salerno, 84084 Fisciano, Italy

**Keywords:** primary school, teaching method, PE, soft skills, transversal skills

## Abstract

Transversal skills are the knowledge, skills, and personal qualities that are currently needed to meet the demands of the working world and everyday life. Schools have the task of equipping students with these skills, working not only on disciplinary goals but also on operational–behavioral goals. In 2018, the European Union adopted new recommendations on eight key competencies for lifelong learning and asked schools to implement new methods to develop these recommendations. To be successful, it is necessary to stimulate students’ development of these competences, which are also called soft skills, from the earliest years of the school experience. Physical education (PE) is called upon to make its contribution. In Italy, the two teaching methods used during PE classes are prescriptive teaching and heuristic learning. It is not clear which of the two methods is the most effective in improving soft skills, especially the skills involved in teamwork. The objective of this article was to compare the effects of these two teaching methods on students’ teamwork skills during PE classes in primary schools. After verifying the normality of the data, a Student’s *t*-test for dependent samples was performed to assess pre-test and post-test differences in each of two groups, while a Student’s *t*-test for independent samples was performed to compare the two groups after 3 months. Heuristic learning proved to be the most effective method for improving teamwork skills. The results may make an important contribution to future teacher training on the most effective teaching methods for developing students’ soft skills.

## 1. Introduction

Italy’s National Directions [1] and the World Health Organization (WHO) [2] agree that physical activity offers numerous opportunities to promote motor, cognitive, social, cultural, and affective experiences. In Italian primary schools, teachers’ actions are structured according to the National Directions [3], which identify goals for the development of competencies and learning objectives. These goals are organized into four thematic cores:the body and its relationship with space and time;body language as a communicative–expressive modality;play, sport, rules, and fair play; andhealth, wellness, prevention, and safety.

However, these four directions do not take into account the challenge the world is currently facing [4] in countering unemployment, which is one of the main problems of young people. Youth unemployment appears to be related to the lack of soft skills in working environments.

Physical education (PE) can be a useful tool in promoting transversal skills. Transversal skills are the knowledge, skills, and personal qualities that are currently needed to meet the demands of the working world and everyday life [5]. Schools have the task of equipping students with these skills, working not only on disciplinary goals but also on operational–behavioral goals. In 2018, the European Union adopted new recommendations on eight key competencies for lifelong learning. As more jobs are automated, social and civic skills are required to ensure resilience and the ability to adapt to change. Among the eight key competencies is “personal, social and learning competence” which consists of the abilities to reflect on oneself, manage one’s time effectively, work within a team, be resilient, and manage one’s own learning and career. Such skills are called soft skills. They are essential in the workplace [6]. To be successful, it is necessary to stimulate the development students’ soft skills from the earliest years of the school experience. PE is called upon to make its contribution.

Sports and PE have been considered as a favorable tool for the development of life skills, values, and character that can benefit individuals’ personal growth, including interpersonal and intrapersonal skills that are important in later stages of life [7]. In the sports world, we often hear about the importance of teamwork, a soft skill that is also indispensable in the work environment. Teamwork allows people to maximize their abilities by placing them in a group context, working cooperatively with others, and achieving common success. Participation in sports can generate positive behaviors, such as supporting teammates or helping injured opponents, while reducing negative behaviors, such as intimidating other players [8]. According to Bailey [9], PE in schools has the potential to make an important contribution to the development of children’s fundamental physical skills and competencies, which are necessary precursors to participation in later lifestyles and physical sports activities. In addition, if presented appropriately, PE lessons can support the development of social skills and behaviors, self-esteem, and school-friendly attitudes. In addition, under certain circumstances, PE can support students’ academic and cognitive development.

Despite the growing interest in the role of PE and sports in promoting personal and social development, it is important to keep in mind that mere participation in PE and sports lessons does not automatically lead to positive outcomes [10]. The study by Cronin et al. [11] concluded that simply implementing a program based on life and soft skills is not sufficient to achieve positive results. It is the responsibility of PE teachers to create appropriate pedagogical circumstances [12]. PE classes, if well planned and structured, can offer practical opportunities for the improvement of soft skills. The focus must be on the teacher and the didactics of teaching [13]. 

In PE, the teaching method predominantly used in schools is the traditionalist one, based on the prescriptive method, in which the student learns through imitation and repetition of motor gestures [14]. The teacher directs the structure of motor programs and their parameterization: he or she guides the whole process by issuing the orders, commands, timing, sequences, purposes, and goals of activities, and by monitoring and reprogramming them. The tools used are exercises, which can be partial, segmented, simplified, varied, randomized, etc. 

The aim of the traditionalist PE teaching method is stabilizing the motor program and minimizing executive variability through repetition. The limitations of that method include the automation of gestures and minimal cognitive involvement [15]. These limitations are overcome by heuristic learning that is based on ecological-dynamic approach, in which students move themselves to the center of the learning process. A key concept within dynamical systems theory is that of self-organization: in order to arrive at the execution of coordinated movements, we move from randomized movements to organized movement phases, based on the self-organizing properties of the system [16]. The teacher frees the students to explore, using educational practices such as applying techniques borrowed from psychology (brainstorming, cooperative learning, circle time, peer tutoring, and tutorship) and manipulating the environment and game rules. The strengths of this approach are high cognitive engagement, a propensity to develop creativity and life/soft skills, greater autonomy, and flexibility [17].

There is insufficient evidence to determine which of the two methods can best promote the development of soft skills through PE in schools. Therefore, further studies incorporating life skills education within specific contexts are needed [18]. Accordingly, the aim of this study was to compare the effects of the two teaching methods in developing teamwork skills during PE classes in primary schools.

## 2. Materials and Methods

### 2.1. Design and Participants

A quasi-experimental study was conducted for two groups, with a pre-test and post-test design. Convenience sampling was used to recruit 100 children from the fourth and fifth grades of three primary schools in Italy. The children had no experience in acquiring teamwork skills. Then, after we verified that the children had the same skill levels in terms of teamwork, they were assigned to two homogeneous groups: an HEUR-L (heuristic learning) group and a PRES-T (prescriptive teaching) group.

The HEUR-L group (n = 46; 9.4 ± 0.5 years old), consisted of students from two fourth and fifth grade classes; during one hour of PE per week for 3 months, the activities of the students were considered using the heuristic learning method, which is based on ecological-dynamic approach.The PRES-T group (n = 54; 9.5 ± 0.5 years old) consisted of students from two fourth and fifth grade classes; during one hour of PE per week for 3 months, the activities of the students were considered using the prescriptive teaching method, which is based on a cognitive approach.

A sports tutor with a degree in sport sciences, having received training in the use of these two approaches to teaching, travelled to the three primary schools for a sports project that implemented the educational intervention. The tutor was assigned to teach PE once per week for 3 months to eight fourth and fifth grade classes.

### 2.2. Educational Intervention

The general scheme implemented by the sports tutor consisted of three phases:

The general and specific warm-up phase: active mobility and runs with exercises of gradual intensity to raise body temperature; the central phase: two activities in the form of games based on basic motor schemes: basketball, volleyball, and soccer; and the cool-down phase: relaxation and static stretching.

#### 2.2.1. Intervention in the HEUR-L Group

In the HEUR-L group, the warm-up phase was performed in rotation by one/two children who volunteered, based on their own experience or on a card with instructions that were previously prepared by the tutor. Then, the tutor set up routes comprising exercises that could be performed in various ways (e.g., a series of pins placed in a snake shape). Each child could choose how to perform the course. Only in the event of difficulties could the tutor intervene with suggestions, without providing the solutions.

The central phase was preceded by a brainstorming activity on the theme/activity of the day, in which the tutor proposed video tutorials or cards to provide the children with more stimulus. Then, the children had 5 min to organize themselves by establishing teams, roles, pairs, captains, etc. The tutor observed their behavior and made some suggestions via questions. During the activities, the children were self-managing and self-organizing, working through the methodologies of cooperative learning and peer tutoring.

In the cool-down phase, the children were arranged in a circle and took turns doing the relaxation exercises, possibly with the help of the others in the group. In the meantime, they expressed their feelings, both positive and negative (circle time), while the tutor made notes to improve the lessons that followed.

#### 2.2.2. Intervention in the PRES-T Group

In the PRES-T group, the lesson proposed by the tutor closely mirrored the traditional method implemented in schools. In the warm-up phase, the tutor had the children perform the exercises. The tutor prepared various routes and, before they were performed, demonstrated the routes by explaining each exercise step-by-step. In the central phase, the tutor explained the activity to be performed and had the children perform the exercises through partial, segmented, varied, randomized exercise techniques to perfect their execution through repetition. In the cool-down phase, the tutor demonstrated the relaxation exercises and the children performed them.

A sample of the lessons performed for both groups is shown in Table 1.

### 2.3. Data Collection

Pre-test and post-test questionnaires were administered at the same time for the HEUR-L and PRES-T groups. The development of skills required in teamwork was assessed using Cronin and Allen’s life skills scale for sport (LSSS) [19]. The LSSS scale includes eight subscales: teamwork (seven items), goal-setting (seven items), social skills (five items), problem solving and decision making (four items), emotional skills (four items), leadership (eight items), time management (four items), and interpersonal communication (four items). Participants answered on a five-point Likert-type scale ranging from 1 (not at all) to 5 (very much). Only the teamwork subscale (seven items) was considered for the study: “Based on your behaviors, how much do you agree from 1 to 5?”

Item 1: I accept suggestions for improvement from others.Item 2: I help build team/group spirit.Item 3: I work well within a team/group.Item 4: I suggest to my team/group members how they can improve their performance.Item 5: I help another member of my team/group perform a task.Item 6: I change the way I work for the good of the team/group.Item 7: I work with others for the good of my team/group.

### 2.4. Data Analysis

Descriptive statistics were used to summarize the data into a shape of mean value ± standard deviation and answer frequencies, provided in percentages. After verifying the normality of the data with the Shapiro–Wilk test and the homogeneity of variances with Levene’s test, a Student’s *t*-test for independent samples was performed to verify that the two groups started from the same level of teamwork skills. A Student’s *t*-test for dependent samples was used to verify the improvements in each group. Statistical significance was set at *p* ≤ 0.05. Data analyses were performed using Statistical Package for Social Science software (IBM SPSS Statistics for Windows, Version 26.0. Armonk, NY, USA).

## 3. Results

The homogeneity test showed that the two groups did not differ significantly before the educational intervention. The HEUR-L and PRES-T groups were similar with respect to gender distribution, weight, and class. A detailed descriptions of the groups’ characteristics is shown in Table 2.

To determine whether the educational program influenced the participants’ teamwork skills, after 3 months we compared the mean pre-test and post-test differences between the two groups. The mean differences and the standard deviations of the pre-test and post-test teamwork of the two groups are presented in Table 3.

Before starting the educational program, the difference between the two groups was not statistically significant (*p* > 0.05). After 3 months, children in the HEUR-L group showed a significant difference (*p* < 0.05) with respect to the PRES-T group for Item 1 [t(98) = 3.32, *p* = 0.001], Item 3 [t(98) = 1.73, *p* = 0.027], Item 4 [t(98) = 2.86, *p* = 0.005], Item 6 [t(98) = 2.22, *p* = 0.029], and Item 7 [t(98) = 2.62, *p* = 0.010]. A detailed description is shown in Table 4.

Both groups improved their teamwork skills (*p* < 0.05), but in different ways. The HEUR-L group improved in all items of the questionnaire, while the PRES-T group improved only in Items 4 and 5. A detailed description is shown in Table 5.

## 4. Discussion

The purpose of this study was to compare two different teaching methodologies implemented during PE lessons, in order to improve teamwork skills of primary school children in the fourth and fifth grades. The main results showed that the HEUR-L group improved significantly in teamwork skills, compared with the PRES-T group, during the 3 month period.

At the beginning, both groups started with the same level of teamwork skills. The lowest values concerned the ability to accept suggestions from peers to improve oneself (Item 1), the ability to suggest to peers how to improve one’s performance (Item 4), and the ability to change one’s way of working for the good of the group (Item 6). Very often, the children tended to want to overcome others [20], without listening to suggestions from the teacher or peers. Therefore, they needed to experience and compare themselves with others in order to understand how they could improve, through practice and observation, while learning the benefits of working together. At the end of 3 months, the gap between the two groups became significantly wider, except for two items: Item 2 was concerned with the ability to help build team spirit; Item 5 was concerned with the ability to help another group member perform a task. Neither methodology was effective in improving these two parameters significantly, although there were slight improvements. Accordingly, with more time and experience, children could increasingly improve these items.

When children have little experience in sports and movement, it is not easy to suggest to others how best to perform a task. Children today move too little, both qualitatively and quantitatively [21]. At school, on the other hand, if there is no PE teacher, PE lessons are not always available except in a piecemeal way by teachers from other disciplines [22]. Many such teachers, however, do not feel adequately trained to carry out PE lessons and, therefore, PE requires a specialized figure in the field, i.e., a PE graduate, to better plan and structure the activities that are to be performed [21].

Analyzing the improvements of each group, we can see that for the HEUR-L group, all values increased significantly. The most improved values concerned the responses to Item 1, accepting the suggestions from others to improve, Item 3, working well within the group, Item 4, suggesting to other group members how to improve one’s performance, and Item 6, changing one’s way of working for the good of the group. Through cooperative learning, with the children’s complete autonomy to manage the didactical tasks proposed by the teacher, the HEUR-L group was able to improve these skills to build a united and cooperative group. Cooperative learning is a student-centered method that allows a class to collaborate and interact in small groups, which is very useful in improving children’s misconduct [23].

Cooperative learning can be attributed to a productive teaching style, as it allows students to elaborate on some directions prepared by a teacher. The teacher plays a crucial role, because he or she is responsible for preparing the necessary materials for the students to use, such as video tutorials projected via an interactive whiteboard or paper diagrams. Implementing cooperative learning takes time and resources. Teachers do not often use the reproductive style, precisely because of the large amount of time required, the lack of adequate training, and/or the difficulty in managing a class in dynamic activities [24,25]. Teachers prefer to propose exercises that are already structured in the command style. 

For the PRES-T group, which used a reproductive style, there were no significant improvements, except for two items. Item 4 was focused on the ability to suggest to the group how to improve their own performances, such as during volleyball S3 games, in order to win and prevail over the other team. Item 5 was concerned with the ability to help a teammate to improve the performance of the task in order to win over the others. The primary goal of the PRES-T group seemed to be in improving themselves and their teammates in order to win, without working on team spirit. This happened because the activities were proposed simply by working on the repetition of movements in order to perfect the motor program, without a specific intervention on social interactions, which do not improve automatically if not properly stimulated.

We need to work specifically on listening, mutually helping, and learning from and to peers, as was the case with the HEUR-L group with heuristic learning. Sport, through these pedagogical interventions, can have a significant impact in helping children to develop the winning aspects of a team, such as the bond between players and the ability to communicate and work together for the same goal [26], and to overcome life traumas, such as the current COVID-19 pandemic [27,28].

This study has some limitations, including the use of convenient samples, resulting in an inability to generalize the results to the entire population, and the absence of evidence on the use of the “teamwork” subscale in isolation from other factors. The choice to isolate this subscale was due to the fact that the protocol aimed to work strictly on group work skills, rather than on other social skills. Furthermore, as the original version of the LSSS was too long, the children’s attention span in answering may have been decreased. Future studies could extend the research by investigating other social skills, beyond teamwork.

## 5. Conclusions

Heuristic learning proved to be the most effective method for improving teamwork skills in primary school children. That method offered a more comprehensive view of how to organize PE lessons to stimulate teamwork skills, due to the emergence of spontaneous solutions, the development of problem-solving skills, and the development of other fundamental life skills, while improving the children’s physical fitness. The results of the present study can make a sensitive contribution to the training of future teachers on the most effective teaching methods for improving students’ teamwork skills during PE lessons in primary school.

## Figures and Tables

**Table 1 ijerph-19-12596-t001:** Example of the activity performed using two teaching methods by the HEUR-L and PRES-T groups.

	HEUR-L Group	PRES-T Group
**General and specific warm-up phase** **(20 min)**	After an initial brainstorming on the exercises that could be performed for a proper warm-up, the children formed a circle and alternated (one per lesson) in playing the role of coaches, offering some sequences of exercises to warm up. The PE teacher set up half motor circuits with sports equipment (cones, chines, balls of different size/texture, obstacles), including propaedeutic exercises for soccer, volleyball, and basketball. The teacher’s task was limited to specifying the starting point and the finishing point.	The PE teacher positioned the children in a circle to perform some sequences of exercises to warm up. The children imitated what the teacher demonstrated. The PE teacher set up half motor circuits with sports equipment (cones, chines, balls of different size/consistency, obstacles), including propaedeutic exercises for soccer, volleyball, and basketball. The teacher’s task was to explain and demonstrate the individual stations (by blocks and then by linking them) before the children’s performance.
**Central phase (30 min)**	The PE teacher sets up two activities of 15 min each. Two volunteer children played the role of captain; they were responsible for choosing and managing their team.	The PE teacher set up two activities of 15 min each, dividing the children into two teams and choosing two captains.
	Game 1: basketball Aim: to work on dribbling and shooting in basketball; decision making under time pressure. (1) Brainstorming on the game of tris and basketball fundamentals; (2) Video tutorial on the game of tris through an interactive whiteboard; (3) The children assembled a motor path to start playing. Game 2: volleyball S3 Aim: to apply the rotation rule in volleyball. (1) Brainstorming on the general rules for playing volleyball; (2) Video tutorial on the rotation rule in volleyball through an interactive whiteboard; (3) The children set up the court by assembling the net and marking its boundaries with cones. Then, two captains placed their players in the six areas. Once the game started, at the appropriate time, the children tried to apply the rule that was previously observed. The PE teacher was limited to the roles of observing, providing some suggestions, and, in case of a mistakes, asking the children for reasons and never providing the solutions. The questions asked included: “Which team did service?” and “Which direction do you rotate?”	Game 1: basketball The PE teacher set up and explained the game for the children to perform. The game was divided by blocks to facilitate understanding: dribbling to the basket, shooting at the basket, running to tris patterns, and inserting the cone. Game 2: volleyball S3 The PE teacher, after setting up the volleyball court, explained the rotation rule. Before starting the game, the teacher had one team practice at a time so that everyone memorized the rule. When it came time to rotate, the teacher gave the command so that the children could execute it.
**Cool-down phase (10 min)**	Circle time: the children formed a circle and, in turns, explained their reflections on the content of the lessons.	Relaxing exercises were demonstrated by the PE teacher.

**Table 2 ijerph-19-12596-t002:** Groups’ characteristics.

	HEUR-L Group	PRES-T Group
	Participants	Participants
Gender		
Male	23 (50%)	26 (48.1%)
Female	23 (50%)	28 (51.9%)
Weight		
Overweight	8 (17.4%)	8 (14.8%)
Normal	33 (71.7%)	42 (77.8%)
Underweight	5 (10.9%)	4 (7.4%)
Class		
Fourth (9 years old)	24 (52.2%)	28 (51.9%)
Fifth (10 years old)	22 (47.8%)	26 (48.1%)

**Table 3 ijerph-19-12596-t003:** Teamwork skills values, pre-test and post-test, after 3 months.

Entry					Exit			
	HEUR-L Group		PRES-T Group		HEUR-L Group		PRES-T Group	
	Mean	SD	Mean	SD	Mean	SD	Mean	SD
Item 1	3.6522	1.40186	3.2778	1.27999	4.0435	0.98785	3.2963	1.22289
Item 2	3.9565	1.11468	3.8519	1.13947	4.1957	0.98024	3.8333	1.09458
Item 3	3.8696	0.85916	3.7963	0.95916	4.1739	0.79734	3.7778	0.94503
Item 4	3.5435	1.27726	3.4630	1.28435	4.1957	0.93380	3.5741	1.19119
Item 5	3.8696	1.25802	3.8148	1.28964	4.1739	0.99564	4.0926	1.01440
Item 6	3.5217	1.41011	3.3519	1.45573	3.9348	1.12353	3.3704	1.37767
Item 7	4.3043	0.96309	4.2963	1.10964	4.5870	0.65238	4.0741	1.17925

**Table 4 ijerph-19-12596-t004:** Comparison between the two groups after the educational intervention.

Student’s *t* Test for Independent Samples After 3 Months							
	*t*	Df	Sig. (Two Tails)	Difference in Mean	Std. Error Mean	95% CI	
						Lower	Upper
Item 1	3.322	98	0.001	0.74718	0.22494	0.30079	1.19357
Item 2	1.730	98	0.087	0.36232	0.20940	−0.05323	0.77786
Item 3	2.243	98	0.027	0.39614	0.17663	0.04563	0.74664
Item 4	2.867	98	0.005	0.62158	0.21682	0.19130	1.05186
Item 5	0.403	98	0.688	0.08132	0.20181	−0.31917	0.48181
Item 6	2.220	98	0.029	0.56441	0.25428	0.05981	1.06902
Item 7	2.626	98	0.010	0.51288	0.19531	0.12530	0.90046

**Table 5 ijerph-19-12596-t005:** Improvement of teamwork skills in each group.

Paired *t*-Test	HEUR-L Group			PRES-T Group		
	*t*	df	Sig. (Two Tails)	*t*	df	Sig. (Two Tails)
Item 1	−3.564	45	0.001	−0.444	53	0.659
Item 2	−1.974	45	0.055	0.256	53	0.799
Item 3	−3.288	45	0.002	0.375	53	0.709
Item 4	−4.354	45	0.000	−2.574	53	0.013
Item 5	−2.047	45	0.046	−2.171	53	0.034
Item 6	−2.737	45	0.009	−0.207	53	0.837
Item 7	−2.227	45	0.031	1.766	53	0.083

## Data Availability

Not applicable.

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
