# Peer review of "Heuristic Learning as a Method for Improving Students’ Teamwork Skills in Physical Education"

_ijerph, 2022, doi:10.3390/ijerph191912596_

Round 1
Reviewer 1 Report (Previous Reviewer 2)
Having read the revised manuscript and the response letter, I realize the extensive revision that has been carried out, for which I commend the authors. Overall I feel that my concerns have been addressed and that the authors provided thoughtful responses to my queries and suggestions. I have no further recommendations for how to strengthen this manuscript.
Author Response
Dear Reviewer,
Thank you for your time. I appreciate it.
Reviewer 2 Report (New Reviewer)
Dear autors,
Manuscript is created to improve teachers' intentions preparing school children in using their skills in everyday life. There are plenty of hardly understanding parts that are bordering with something that is generally false. Like as your manuscript is not written for someone else reading purpose. That, I can't recommend your manuscript to be published in that moment.
Moreover, there are highlighting text without any reason. Please send me later a manuscript without any intervention.
The abstract is not such a good place to introduce experiment technics details. Punctuations are validable part of the text writing serving for making reading easier. At the same time, highlighting offends readers.
Obviously you know how to write a good manuscript and there is no reason that you would not create a better version for the next review. Furthermore, readers will be inspired with your teaching ideas. Clumsiness in writing ruins your valuable work.

Author Response
Dear Reviewer,
thank you for your time. I appreciate it. I enclose the pdf with your comments and my point-by-point replies. The changes performed can be found in the final uploaded version of the manuscript using word-tracking.

This manuscript is a resubmission of an earlier submission. The following is a list of the peer review reports and author responses from that submission.
Round 1
Reviewer 1 Report
The introduction needs to be edited and updated since it is hard to follow.
It is very vague how PE classes with heuristic learning could promote students' overall teamwork skills.
It is hard to find how the authors embed transversal skills into activities. Sample activities in table 1 do not support enough how the authors implemented transversal skills into the activities. It is hard to find uniqueness of activity design.
There is no indication how the authors controlled the participants who already have had previous experiences for heuristic learning/transversal skills.
Reviewer 2 Report
The manuscript “Heuristic learning as a method to improve teamwork skills in physical education” reports a quasi-experimental study held in Italy, aiming to compare the effects of two teaching and learning methods regarding the development of teamwork skills during physical education lessons in 4th and 5th graders.
The paper presents all sections expected from a scientific report, the content in each section is globally adequate, and the writing is clear.
The first section of the manuscript presents the rationale for this study, also showing little evidence regarding the best approach to develop teamwork skills in PE lessons. I recommend the authors carry out some minor revisions, as follows:
- On page 2, line 6, one reads “to implement new methods to develop students at 360°”, but what does that mean exactly? Without further information, the reader needs to check the Recommendation of the Council of the European Union, from 2018. But where can it be found? So maybe the authors can complement this idea, offering more details and a reference.
- On line 66, “et al” should be “et al.”
- On table 1, the authors present an “example of the activity performed using two teaching methods by EXP and CON group”, which is quite helpful in understanding the differences made in the teachers’ pedagogy. However, the reader knows little about the remaining lesions of the experiment that lasted for 3 months. It is possible to present more information regarding the content of the other lessons? A summary table would suffice, showing the overall experiment.
- As nothing is said about the teacher or teachers who implemented the lessons, either on the EXP or the CON group, the authors should include some details in this regard. How did the teacher develop expertise in the heuristic learning method, “based on the ecological-dynamic approach”? And did the same teacher teach the two groups of students? This is especially relevant as in the discussion it is said that, at this level, there is no PE teacher in Italy and that the lessons are taught by a teacher from another subject (line 180).
The materials and methods section is also clearly presented and every step of the statistical analysis is conveniently justified. On lines 98-99, it is said “Then, after an initial assessment, they were randomly assigned into two groups: EXP (experimental) and CON (control) group.” Maybe the authors could explain what the initial assessment consisted of and what its purpose was. Also, table 2 does not read well as its heading is left at the end of page 5. The items of the teamwork subscale are presented using a numbered list – 1. Item 1accept suggestions… - but I wonder if it could work best using bullets; as it is, it seems redundant.
The results section is concise but still presents evidence that, despite the initial homogeneity of the two groups, after the intervention, the experimental group improved their teamwork skills, in every item of the questionnaire.
The discussion section is the one that needs more improvements, especially because the authors make some claims in interpreting their results that are not empirically nor theoretically supported. For instance, the authors say:
- “Very often children tend to want to excel over others, without listening to suggestions from the teacher or peers. They, therefore, need to experience and compare themselves with others in order to understand through practice and observation, how they can improve.” (Ln 168-171). These claims seem common sense based, but what is the scientific motivation for them? Please provide some reference on this respect.
- “Neither methodology was effective in improving this parameter, probably because it was not an easy thing to achieve when dealing with multiple children with different personalities.” – Again, is this supported by the literature? Please provide some reference in this respect.
The paper needs revising regarding punctuation.
Overall, the research presented in this manuscript was soundly designed and implemented, and consistently reported. Although the authors focus on a short section of their data, their results seem interesting and could inform future training of future teachers on the most effective methods to improve teamwork skills in PE at the primary school level.